# Social Frailty and Social Isolation in Chinese Community-Dwelling Older Adults: A Network Analysis

**DOI:** 10.3390/nursrep15090315

**Published:** 2025-08-27

**Authors:** Hai-Yan He, Di-Fei Duan, Lin-Jia Yan, Lin Lin

**Affiliations:** 1School of Nursing, Sichuan College of Traditional Chinese Medicine, No. 1, Jiaoyu Middle Road, Fucheng District, Mianyang 621000, China; hehaiyan@scctcm.edu.cn; 2Department of Nephrology, Kidney Research Institute, West China Hospital of Sichuan University, Chengdu 610041, China; duandifei89@wchscu.cn; 3The Nethersole School of Nursing, Faculty of Medicine, The Chinese University of Hong Kong, Hong Kong, China; 1155185244@link.cuhk.edu.hk

**Keywords:** social frailty, social isolation, older adults, network analysis, nursing

## Abstract

**Background:** China’s rapidly ageing population faces a double burden of social frailty (SF) and social isolation (SI), both of which accelerate functional decline and increase healthcare use. Clarifying their interplay is essential for nurses, who are often the first to assess and intervene in older adults’ social health. **Method:** In a cross-sectional study, 451 community-dwelling adults aged ≥ 60 years (median = 71) completed the HALFT Social Frailty Scale and the Social Isolation Scale for Older Adults. A mixed graphical model based on Spearman correlations mapped symptom-level associations between SF and SI and identified central nodes. **Result:** SF was present in 22.8% of participants, and the median SI score was 13 (IQR = 9–16). The strongest edge linked “inability to help others” (SF1) with reduced “face-to-face contact frequency” (SI1) (edge weight = 1.85). Central nodes were “lack of social participation” (SI2), “sense of belonging” (SI4), and “lack of someone to talk to” (SF5), indicating key points where SF and SI converge. **Conclusions:** The tight network connecting SF and SI suggests that nursing assessments should screen for both constructs simultaneously. Interventions that increase social participation, foster belonging, and create opportunities for reciprocal helping may mitigate both syndromes, supporting healthier ageing and reducing downstream healthcare utilization.

## 1. Introduction

Global population ageing is accelerating: by 2050, the number of people aged 60 years and older is projected to reach ≈ 2.1 billion worldwide [1]. As the world’s most populous country, China faces an even steeper shift—its 2020 census recorded 18.7% of the population aged ≥60 years (264.0 million people), and UN projections indicate this share will rise to ≈38.8% by 2050 [2,3]. Hence, prioritizing the well-being of older adults is essential to enhance their quality of life and support healthy, active aging. However, despite efforts to improve healthcare, long-term care services in China remain inadequate, making it critical to address these gaps in care to meet the needs of the aging population [4,5].

Within this context, social frailty—a multidimensional construct encompassing reduced social participation, weakened support networks, and diminished community engagement—has emerged as a critical determinant of adverse health outcomes among older adults [6]. Among community-dwelling older adults, the prevalence of social frailty is approximately 18.8% [7]. Recent research indicates that social frailty is widespread in China, with a national survey revealing that approximately 22% of older adults experience moderate to high levels of social frailty [8]. This finding is especially concerning given the strong association between social frailty and negative health outcomes, including increased risks of hospitalization, functional decline, and mortality [9,10].

Social isolation, defined as an objective lack of social connections or interactions, has been recognized as a pervasive issue among older adults [11]. A recent study based on data from the Chinese Longitudinal Healthy Longevity Survey (CLHLS) reported that about 34.8% of community-dwelling elders experience significant social isolation [12]. This is further exacerbated by the unique cultural and demographic context of China, where traditional family support systems are increasingly strained due to urbanization, migration, and the one-child policy [13]. The consequences of social isolation are profound [11], and it has significant health consequences for older adults, including increased risks of depression, cognitive decline, and various chronic diseases such as dementia, heart disease, stroke, and cancer [14,15]. It also slows recovery from illness and contributes to higher rates of comorbidities, disability, and rehospitalization. Ultimately, social isolation is a major factor in premature death and loneliness among older adults [16,17].

These challenges are particularly acute in community settings, where older adults often experience fragmented access to resources and limited opportunities for social engagement [18]. The combination of social frailty and isolation creates a vicious cycle that negatively impacts health and well-being, placing increasing pressure on healthcare and social support systems. While social frailty and isolation are distinct concepts, their interplay is becoming more apparent [19]. Existing research highlights the compounded effects of physical and cognitive frailty in older populations, with strong associations with falls, hospitalization, and mortality [20]. More recently, attention has shifted to social frailty, which can both predispose individuals to isolation and be exacerbated by it, reinforcing the cycle [21,22]. Recognizing the bidirectional link between social frailty and isolation, community and primary care nurses should screen for both conditions during routine assessments and then deploy evidence-based measures—such as structured social participation or peer-support programs and social prescribing that connects patients with community resources—to foster engagement and a sense of belonging [23,24]. However, current evidence often treats social frailty and isolation as separate entities. Network analysis, a methodological approach that maps complex relationships between variables as interconnected nodes, offers a novel way to untangle these associations. Previous studies have used network models to explore symptom clusters in mental health and chronic diseases, identifying central nodes that drive systemic dysfunction [25,26]. When applied to social health, this approach could uncover core components that mediate the frailty–isolation relationship, providing insights for more targeted interventions [27,28].

Despite the increasing recognition of social frailty and isolation among older adults, no study has systematically examined the structural relationships between these two factors in Chinese older adults using network analysis. This gap hinders the development of culturally appropriate strategies to mitigate their combined impact. The present study aims to (1) construct a network model to visualize and quantify the interactions between the components of social frailty and social isolation and (2) identify central nodes and bridging pathways that exacerbate vulnerability. By integrating theoretical insights with practical applications, this research seeks to contribute to both the academic understanding and the formulation of effective strategies to address the dual challenges of social frailty and isolation in aging populations.

## 2. Materials and Methods

We used the STROBE Checklist for a more rigorous study design and improved article quality.

### 2.1. Study Design and Participants

This was a cross-sectional study that involved the recruitment of 451 older participants between September and December 2024 using a convenience sampling approach. Participants were recruited from two community settings in southwestern China—Chongqing (a centrally administered municipality with ~31.9 million permanent residents; urbanization 71.67%; ~387 persons/km^2^; ≥60 years = 21.87%) and Luzhou, Sichuan (a prefecture-level city with ~4.271 million permanent residents; urbanization 53.8%; ~349 persons/km^2^; ≥60 years = 22.99%)—both with established health care access (Chongqing: 23,389 medical and health institutions, 255,200 beds; ~3.2 physicians, 4.0 nurses, 8.0 beds per 1000 residents; Luzhou: 4520 institutions, 39,369 beds; ~3.35 physicians, 4.15 nurses, 9.22 beds per 1000), providing relevant urban and mixed urban–rural contexts for community-dwelling older adults [29,30,31].

The inclusion criteria required participants to be aged 60 or older, without severe impairments in hearing, speech, or cognition that would hinder communication, who voluntarily consented to participate. We defined ‘older adults’ as age ≥60 years, consistent with the People’s Republic of China’s legal definition of the elderly and international usage by the United Nations/World Health Organization [1,32]. This threshold also captures adults aged 60–64, an early-old group relevant to healthy ageing and the onset of social frailty [8,33]. Those excluded from the study included individuals with severe acute or chronic conditions (e.g., terminal illness, advanced cancer) that would interfere with participation, or those who were unable to complete the study procedures independently or with help from a research assistant (RA). To ensure data quality and reliability, several quality control measures were implemented. The research team trained three RAs to standardize data collection and reduce bias. A structured protocol was followed for consistent participant recruitment, and regular monitoring was conducted to verify data accuracy and participant compliance. These measures ensured the scientific rigor and reliability of the study results. Further details of the recruitment process can be found in Appendix A.

### 2.2. Sample Size Calculations

We determined our sample size a priori using two complementary approaches and adopted the larger estimate. First, for estimating the prevalence of social frailty, we applied a precision-based calculation (PASS 15.0, Proportion module), specifying *p* = 0.22 based on prior data, 95% confidence, and a margin of error (half-width) of 5.5%, which yielded a required sample size of n = 418 [8]. Second, our main analytical method—a partial correlation network with 11 items—included 66 parameters (55 pairwise edges + 11 node intercepts) [34]. We followed conservative structural equation modeling guidance recommending at least 5 cases per estimated parameter, consistent with Bentler & Chou’s rule of thumb, yielding n ≈ 330 [35]. Because larger samples improve the stability and accuracy of network estimates (e.g., edge weight precision and centrality reliability) [34], we selected n = 418 to satisfy both methodological needs.

### 2.3. Measurements

#### 2.3.1. Demographic Data and Covariates

The first section of the questionnaire gathers demographic data, including age, gender, marital status, economic status, employment status, educational level, and the presence of chronic diseases, etc.

#### 2.3.2. The HALFT Scale

The scale is made up of one letter from each of the words: Help, Activities, Loneliness, Financial, and Talk. It is a straightforward self-report tool designed to assess social frailty, consisting of 5 items with scores ranging from 0 to 5. A score of 0 indicates no social frailty, scores between 1–2 suggest early-stage social frailty, and scores of 3 or higher indicate social frailty. This scale is commonly used among older adults living in the community, with a Cronbach’s α coefficient of 0.602 [36]. Although the value is relatively modest, it is deemed acceptable for short, formative scales with a limited number of items. The concise design of the HALFT scale, along with its prior validation among community-dwelling older adults in China, supports its practical use—especially in contexts where reducing respondent burden and keeping interviews brief is important [36]. In this study, Crobach’s α was 0.662.

#### 2.3.3. The Social Isolation Scale (SIS) for Older Adults

The SIS for Older Adults, developed by Dr. Nicholson and colleagues in 2019, assesses social isolation in older adults over the past month from both objective and subjective perspectives [37]. The scale includes six items: three on contact (e.g., frequency of contact with family, friends, and neighbors) and three on belonging (e.g., perception of social isolation). Both dimensions use a Likert scale ranging from 0 to 4, with the fifth item scored in reverse. The total score ranges from 0 to 24, with lower scores indicating higher social isolation risk. Cronbach’s α coefficient is 0.77. The Chinese version of the scale is 0.763 [38].

#### 2.3.4. Statistical Analysis

SPSS 26.0 software was used to calculate the means, standard deviations, and Cronbach’s α coefficients for social frailty and social isolation. R 4.1.1 software was used for network model construction.

Network model construction: Because social frailty and social isolation are multidimensional and interdependent, we applied psychometric network analysis to estimate conditional item–item associations without assuming a single latent construct, to identify central and bridging nodes as potential intervention targets, and to evaluate the accuracy and stability of the network structure using nonparametric bootstrapping [39]. We used the R package (version 4.4.1) qgraph to construct a Mixed Graphical Model (MGM) of the social frailty–social isolation network in older adults [34,40]. The Spearman rho correlation method was used in the network construction. The nodes in the network represented items of social frailty and social isolation. The correlations of symptoms were represented by edges, and the calculation of the correlation between two nodes was conducted after statistical control for the influence of all other nodes included in the network [41]. Edge colors encode the sign of the partial association: positive edges are shown in blue and negative edges in red. Thicker edges represent higher correlations. The Fruchterman–Reingold algorithm was used to arrange the network layout; in this layout, strong correlations are placed in the center of the network, and weak correlations are placed in the periphery of the network [41].

We used the R package bootnet to test the significance of edge weight differences between node pairs and evaluate their accuracy. Bootstrapping (1000 samples, α = 0.05) was applied to test the differences, and the 95% confidence interval was estimated to assess accuracy. A narrow confidence interval indicates high accuracy, as per Epskamp et al. [34].

## 3. Results

### 3.1. The Characteristics of Participants

This study included 451 community-dwelling older adults, with a median age of 71 years (IQR: 66–80). The sample was nearly balanced by gender, with 49.9% male. Education levels varied: 37% had ≤6 years of formal education, 50.3% had 6–12 years, and 12.7% had >12 years. Most participants were married (73.3%) (see Table 1).

### 3.2. Social Frailty Severity and Social Isolation Score of Participants

There were 164 participants without social frailty, accounting for 36.4% of the sample. A total of 184 participants (40.8%) were classified in the pre-frail stage of social frailty, while 103 participants (22.8%) were identified as frail. The median score for social isolation was 13 (IQR: 9–16). The boxplot for both variables is shown in Figure 1.

### 3.3. The Network Structure of Social Frailty and Social Isolation in Chinese Older Adults

The network analysis identified key relationships between social frailty (SF) and social isolation (SI). Most of the cross-community edges were positive. The edges with the highest edge weights were SF1 “Inability to help others”—SI1 “See face-to-face at least once a month” (edge weight = 1.85), SF2 “Limited social participation”—SI3 “Feel close on a personal level” (edge weight = 0.78), and SF3 “Loneliness”—SI2 “Communicate on a personal level” (edge weight = 0.68). It is worth mentioning that there were 2 negative cross-community edges: SF1 “Inability to help others”—SI2 “Communicate on a personal level” (edge weight = −0.23) and SF2 “Limited social participation”—SI3 “Feel close on a personal level” (edge weight = −0.23). Appendix A in the Appendix A provides additional details on the network structure. The analysis revealed that the nodes with the highest Strength values were SI2, SI4, and SF5. These nodes were identified as having the most significant direct influence within the network, indicating their central role in the connectivity structure (see Figure 2).

As depicted in Appendix A of the Appendix A, the precision of edge weight estimates is supported by the narrow 95% bootstrapped confidence intervals. The results of the bootstrapped difference test for edge weights can be found in Appendix A of the Appendix A.

## 4. Discussion

This study represents the first application of network analysis to investigate the intricate relationship between social frailty and social isolation among community-dwelling older adults in China. Our findings demonstrated that SF and SI were not only prevalent but also interconnected in the Chinese context, reflecting the unique demographic and cultural challenges facing this population.

In our study, the median score for SI was 13 (IQR: 9–16), indicating a moderate level of social isolation across the sample. For social frailty (SF), 40.8% were pre-frail and 22.8% frail, pointing to a considerable social vulnerability burden. These SF figures align with prior estimates of 20–22% SF prevalence in Chinese older adults [7,12]. At the national level, data from the China Health and Retirement Longitudinal Study (CHARLS) report comparable SI proportions (~35–44%) when using multi-item indices [42]. A 2023 global meta-analysis pooled SI prevalence at 25% (95% CI: 21–30%) [43], indicating that Chinese older adults in our study may face comparatively higher isolation levels. This elevated burden of SF and SI may be influenced by rapid urban transitions and internal migration in China, which can disrupt intergenerational contact and social cohesion [42,44,45]. This high prevalence also underscored the substantial public health challenge posed by these conditions, particularly given their strong association with adverse health outcomes, including increased risks of hospitalization, functional decline, and mortality [10,16]. In the context of ongoing demographic transitions, it is imperative to prioritize interventions that enhance community-based social networks and address systemic gaps in long-term care services. Examples include the expansion of home care services and the development of integrated healthcare models [46]. Evidence suggests that home- and community-based services (HCBSs) in China are associated with markedly lower levels of loneliness among older adults with functional limitations (OR = 0.81; 95% CI: 0.63–0.99) [47]. Moreover, China’s policy shift toward integrated community care—linking primary care, community resources, and social services—aims to reduce service fragmentation while simultaneously fostering personal contact and social participation [48,49].

In addition, the network analysis provides a novel perspective on the interplay between SF and SI, suggesting that they are not merely co-occurring phenomena but are interlinked in a complex relationship. The network analysis revealed complex interactions between SF and SI components. Appendix A highlights several notable correlations. For instance, the edge weight between SF1 (“Inability to help others”) and SF2 (“Limited social participation”) was 1.849, indicating a robust positive association. This aligned with prior findings that reduced social participation often coexists with diminished capacity to provide support, reinforcing frailty [6]. Conversely, SF5 (“Lack of someone to talk to”) and S12 (“Perception of isolation”) exhibited a negative edge weight of −1.108, suggesting that frequent communication may buffer subjective feelings of isolation. This is consistent with a study emphasizing the protective role of social interaction against perceived loneliness [11]. In addition, the edge weight of 1.182 between SI2 (“Communicate on a personal level”) and SI1 (“See face-to-face at least once a month”) signified a robust connection, suggesting that individuals who frequently communicate personally tend to have regular face-to-face interactions. This aligned with a previous study that indicated that maintaining personal communication is crucial for fostering social networks among older adults [50]. Lastly, the edge weight of 1.732 between SI5 (“Belonging”) and SI4 (“Feel close on a personal level”) further emphasizes the importance of emotional connections in reducing feelings of isolation. This high correlation suggested that individuals who feel a sense of belonging are more likely to have close personal relationships, which aligns with a previous study where strong social ties were shown to significantly decrease loneliness and improve overall health outcomes among older adults [51].

The network analysis also identified SI2 (“Personal communication”), SI4 (“Lack of belonging”), and SF5 (“Lack of someone to talk to”) as central nodes within the interconnected system of social isolation and social frailty among community-dwelling older adults. These findings highlight the critical role of relational disconnection and perceived social marginalization in shaping psychosocial vulnerability in aging populations [52,53]. SI2 and SF5 emerged as key pathways through which social frailty exacerbates isolation, with deficits in communication and confidant availability contributing to functional decline and emotional withdrawal [54]. SF5’s centrality suggests that the absence of conversational partners not only reflects social frailty but also perpetuates isolation, consistent with the “disuse atrophy” hypothesis in social gerontology [55]. SI4, reflecting a sense of exclusion, was identified as a crucial determinant linking subjective feelings of detachment from community activities to objective relational deficits, further emphasizing the importance of belongingness in social frailty. The identification of SI2 (personal communication), SF5 (lack of someone to talk to), and SI4 (belonging) as central nodes in the SF–SI network suggests clear entry points for public health strategies. Routine dual screening for social frailty and isolation within primary-care settings—aligned with the WHO’s ICOPE framework—can systematically flag at-risk individuals [56]. Building on this, nurse-facilitated interventions, such as peer companionship, intergenerational programs, and social prescribing to community assets (e.g., cultural, volunteer, exercise groups), may be particularly effective in restoring communication, confidant access, and belonging [57]. These models are highly compatible with China’s national priorities, including the Integrated Health and Elderly Care model and the Healthy China 2030 plan [58], both aiming to promote healthy ageing through community-based, person-centered services.

Despite the valuable insights gained from this study, several limitations must be acknowledged. The cross-sectional design precludes the establishment of causal relationships between social frailty and social isolation. The convenience sampling methodology might have introduced sampling bias, potentially limiting the generalizability of the findings to other settings. The study’s focus solely on the perspectives of older adults might limit a comprehensive understanding of the complex dynamics of social health; including the views of family members, healthcare providers, and other relevant stakeholders would significantly enrich future research. Our study defined older adults as ≥60 years, but the lack of age-stratified analyses (e.g., 60–64 vs. ≥65) may limit comparability with studies using a 65-year cut-off; future work should refine SI and SF analyses across age segments to address this gap.

### Implications for Clinical Nursing Practice

The findings of this study carry important implications for community and primary care nursing practice. Integrating dual screening for social frailty and social isolation into comprehensive geriatric assessments is warranted, as both conditions were highly prevalent and interrelated in our sample, underscoring the need for early detection [59]. Network analysis further revealed that personal communication (SI2), lack of confidants (SF5), and belonging (SI4) function as central nodes in the SF–SI interplay. Existing evidence indicates that structured group activities (e.g., tai-chi, walking clubs, arts programs), peer companionship, and social prescribing to community resources (e.g., volunteering, exercise, educational groups) can reduce loneliness, enhance connectedness, and foster a sense of belonging [60,61,62]. These strategies directly target the central nodes identified in our analysis and hold promise for mitigating SF and SI. Evidence from systematic reviews further supports the effectiveness of such social engagement approaches in reducing loneliness and restoring connectivity, thereby weakening the SF–SI cycle and improving health outcomes in older adults while easing pressure on healthcare systems [63]. Although not all of these interventions are nurse-led, nurses are uniquely positioned within community and primary care teams to adapt, implement, and evaluate them [64]. Encouraging the development and validation of nurse-led models based on existing evidence may provide scalable solutions to support healthy ageing in this population.

## 5. Conclusions

This study highlights the intertwined nature of social frailty and isolation among Chinese older adults, identifying communication and perceived isolation as critical intervention targets. Culturally adapted community programs addressing reciprocal support and subjective loneliness could mitigate adverse health outcomes, offering a roadmap for policymakers and healthcare providers to enhance aging-in-place initiatives.

## Figures and Tables

**Figure 1 nursrep-15-00315-f001:**
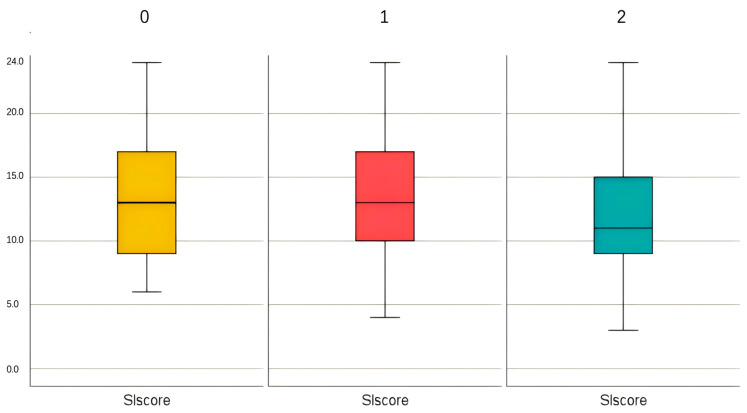
Boxplot of Social Frailty Severity and Social Isolation Score. 0 = without social frailty; 1 = with the pre-frail stage of social frailty; 2 = with social frailty; SI = social isolation.

**Figure 2 nursrep-15-00315-f002:**
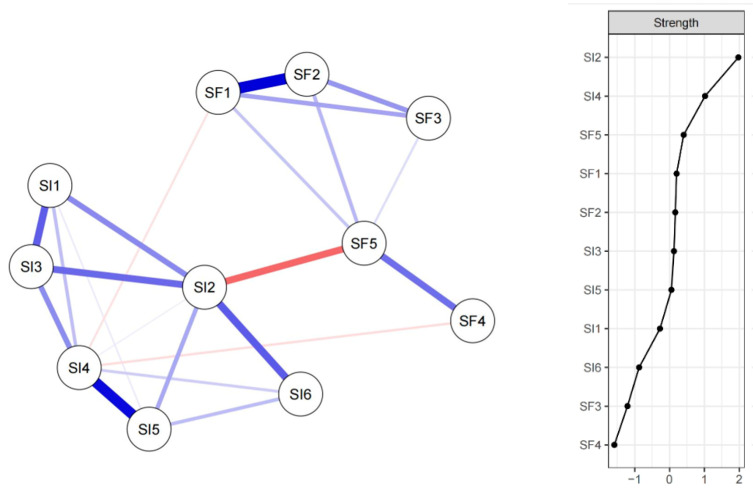
The network structure of social frailty and social isolation in Chinese older adults. Note: SF1 = Inability to help others; SF2 = Limited social participation; SF3 = Loneliness; SF4 = Economic hardship; SF5 = Lack of someone to talk to. SI1 = See face-to-face at least once a month; SI2 = Communicate with on a personal level; SI3 = Feel close to on a personal level; SI4 = Feel that my relationships are fulfilling; SI5 = I feel like I just don’t belong; SI6 = Spend enough time in social activities; Edges are colored by sign (blue = positive, red = negative), and their width/saturation indicates absolute edge weight.

**Table 1 nursrep-15-00315-t001:** The characteristics of participants (n = 451).

Variables	Median, IQR, Percentage (n = 451)
Age	71 (66, 80)
Gender	
Male	225 (49.9)
Female	226 (50.1)
Educational years	
≤6 years	167 (37.0)
>6 to 12 years	227 (50.3)
>12 years	57 (12.7)
Marital status	
Married	331 (73.3)
Divorced/Widowed	120 (26.7)
Source of income	
Children	120 (26.7)
Pension	283 (62.7)
Other	48 (10.6)
Household Per Capita Monthly Income	
≤2000 yuan	72 (15.9)
>2000 to 4000 yuan	147 (32.5)
>4000 yuan	75 (16.6)
Not reported	157 (34.8)
Employment status	
Retired/Unemployed	431 (95.5)
Employed	20 (4.5)
Number of children	
≤1	173 (38.3)
2	148 (32.8)
≥3	130 (28.8)
Living alone	
Yes	117 (25.9)
No	334 (74.1)

## Data Availability

The datasets analyzed in this study are available from the corresponding author upon reasonable request.

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
