# Peer review of "Social Frailty and Social Isolation in Chinese Community-Dwelling Older Adults: A Network Analysis"

_nursrep, 2025, doi:10.3390/nursrep15090315_

Round 1

Reviewer 1 Report

Comments and Suggestions for Authors

Journal: Nursing Reports
Title: The Social Frailty and Social Isolation in Chinese Community-dwelling Older Adults: A Network Analysis
Authors: Hai-yan He, Di-fei Duan, Lin-jia Yan, Lin Lin

Recommendation: Minor Revisions Suggested

This manuscript presents a well-structured and methodologically deep approach into the relationship between social frailty and social isolation among Chinese community-dwelling older adults, utilizing network analysis. The topic is timely and relevant, with important implications for gerontological nursing practice and community health policy. The novelty lies in applying a network approach to this intersection, offering actionable insights for intervention development.

Abstract: Clearly states the background, aims, methods, results, and conclusions in a concise way. Second very good point is that highlights the novelty of applying network analysis to the interplay between social frailty and isolation.

There is no need to put numbers in abstract section.

Introduction: is very well written showing urgency and relevance of the topic and new methodology approach.

Methods:

Clear description of sampling, inclusion/exclusion criteria, and quality control measures.I like detailed explanation of network analysis procedures and rationale for using specific statistical packages. STROBE following is correct and sample size calulation well inserted.

The relatively low Cronbach’s α (0.602) for the HALFT scale should be addressed with a rationale for its continued use in this context. Have you measured Cronbach alpha on your sample? Please give further elaboration. 

Results: Comprehensive presentation of participant characteristics, prevalence rates, and network structure findings. Identification of central nodes adds actionable insights for intervention design.

Discussion: Thoughtful integration of findings with prior literature.

The manuscript makes a meaningful contribution to the literature and science on social determinants of health in aging populations. With the above refinements, it will be even stronger in methodological transparency and clarity.

I recommend acceptance after minor revisions.

Author Response

We would like to express our sincere gratitude to the reviewers for your meticulous evaluation and insightful feedback on our manuscript. Their constructive comments and valuable suggestions have significantly contributed to enhancing the clarity, rigor, and overall quality of our work. 
Comments 1: The relatively low Cronbach’s α (0.602) for the HALFT scale should be addressed with a rationale for its continued use in this context. Have you measured Cronbach alpha on your sample? Please give further elaboration. 

Response 1: Thank you for your valuable feedback. We have added supplementary information to further elaborate on this issue.

Line 147: Although the value is relatively modest, it is deemed acceptable for short, formative scales with a limited number of items. The concise design of the HALFT scale, along with its prior validation among community-dwelling older adults in China, supports its practical use—especially in contexts where reducing respondent burden and keeping interviews brief is important[30]. In this study, the Crobach‘s α was 0.662.

Reference:

  1. Ma L, Sun F, Tang Z. Social Frailty Is Associated with Physical Functioning, Cognition, and Depression, and Predicts Mortality. J Nutr Health Aging. 2018;22(8):989-995. doi:10.1007/s12603-018-1054-0

Reviewer 2 Report

Comments and Suggestions for Authors

I appreciate the opportunity to contribute to the peer review process. While the research addresses an important topic and employs an interesting analytical approach, I would like to point out some methodological and presentation issues that should be addressed before publication.

1. Comment on Language and Terminology:

The authors use the term "elderly" the manuscript when referring to the study population. I recommend replacing "elderly" with "older adults" or "older people" to align with current best practices in gerontological research.

The term "elderly" is increasingly considered potentially ageist and may carry negative connotations. Leading geriatric organizations, including the American Geriatrics Society and major gerontology journals, now recommend using "older adults" as it is more respectful and person-centered language.

2. Comment on Figure Axis Setting:

In the Methods section, the authors state that the Social Isolation Scale has a theoretical range of 0-24 points, yet the box plot displays a y-axis extending to 25 points. While the actual data appears to be within the correct theoretical range, this discrepancy between the stated range and the figure presentation may confuse readers.

Please adjust the y-axis of the box plot to reflect the theoretical maximum score of 24 points as described in your Methods section. This will ensure consistency between the methodological description and the data presentation, and provide readers with a more accurate visual representation of the scale's range.

3. Comment on Figure 2:

The network diagram uses different colors for edges, but the meaning of this color coding is not explained in the figure caption. Please provide a clear explanation of what the colors represent.

4. Comment on Study Population and Setting:

The authors conducted convenience sampling in Luzhou city and Chongqing city, but provide insufficient information about the characteristics of these locations. To enhance the generalizability and interpretation of findings, please provide relevant demographic and geographic information about both cities, including:

  • Urban vs. rural classification
  • Population density
  • Aging rate/proportion of older adults
  • Socioeconomic characteristics
  • Healthcare accessibility

This contextual information is crucial for readers to understand the study setting and assess the external validity of the findings.

5. Comment on Age Criteria:

There is an inconsistency between the background discussion and study inclusion criteria. The background section primarily discusses challenges associated with aging among adults aged 65 years and older, which aligns with conventional definitions of older adults. However, the study includes participants aged 60 years and older without adequate justification for this age threshold.

Please provide a clear rationale for:

  • Why 60 years was chosen as the lower age limit
  • How this choice relates to the study objectives
  • Whether this age criterion is appropriate for the Chinese cultural context
  • How this may affect the interpretation of results in relation to existing literature that typically focuses on adults aged 65+

6. Comment on Supp Fig1:

The details of the recruitment process are explained in Supp Fig 1, but please confirm that this is correct.

7. Comment on Sample Size Calculation:

The authors mention that the sample size was increased "to ensure robustness," but this explanation lacks sufficient detail and scientific justification. About 30% increase(N=418) from the calculated sample size (N=330) represents a substantial change that requires clear methodological reasoning.

8. Comment on Discussion:

  • The authors discussed that “the high burden of SF in China may be attributed to its unique sociocultural background, such as urbanization and the strain on the family support system caused by the one-child policy.” I recommend that careful consideration is needed to determine why urbanization and the one-child policy contribute to the high burden of SF in China.
  • The authors mention the need for strengthening community-based social networks and implementing interventions (such as home care services and integrated healthcare services) to address systemic disparities in long-term care services. However, the mechanism by which home care services and integrated healthcare services reduce SI among older adults in China requires better explanation.
  • The authors should better articulate the role of nursing interventions in addressing the identified network connections between social frailty and isolation. Specifically, please provide evidence from previous studies demonstrating how nursing-led interventions (such as structured social participation groups, peer support networks, and social prescriptions) can effectively target the central nodes identified in your network analysis.

Comments on the Quality of English Language

Reviewer 3 Report

Comments and Suggestions for Authors

Dear authors,
I appreciate your efforts to understand the complex interaction between social instability and isolation, which will help improve quality of life and promote healthy ageing.  Given the growing ageing population worldwide, prioritising research in this area is essential for public health planning and policy-making.

Statistical analysis is detailed in the text, including software, network model construction, and accuracy testing.

I propose to briefly add a justification for the use of the network analysis method in the context of social instability and isolation in the "Statistical analysis" section.

In the "Discussion" section, I propose to use your results in the context of the public health implications.

The proposed manuscript titled “The Social Frailty and Social Isolation in Chinese Community-dwelling Older Adults: A Network Analysis” is well written and addresses an important question that is suitable for publication. 

However, the protocol would need several revisions to improve readability. Below are specific suggestions for improvement:  

1.The network analysis identified several key relationships between social frailty and social isolation in the study sample. The use method of network analysis in the context of social frailty and isolation is not clearly stated in the article. A brief justification would strengthen the methodological section.  

2. In the Discussion section, please consider using the results of your findings with public health implications.  

Finally, I would like to commend the authors for their efforts to understand the complex interaction between social instability and isolation. Given the increasing aging population worldwide, prioritizing research in this area is essential for public health planning and policymaking.

Author Response

Comments 1: However, the protocol would need several revisions to improve readability. Below are specific suggestions for improvement:  

  1. The network analysis identified several key relationships between social frailty and social isolation in the study sample. The use method of network analysis in the context of social frailty and isolation is not clearly stated in the article. A brief justification would strengthen the methodological section.  

Response 1: Thank you for your valuable comment. We have added a brief justification in the Methods section.

Line 166: Because social frailty and social isolation are multidimensional and interdependent, we applied psychometric network analysis to estimate conditional item–item associations without assuming a single latent construct, to identify central and bridging nodes as potential intervention targets, and to evaluate the accuracy and stability of the network structure using nonparametric bootstrapping[33].

Reference:

  1. Borsboom, D., Deserno, M.K., Rhemtulla, M. et al. Network analysis of multivariate data in psychological science. Nat Rev Methods Primers 1, 58 (2021). https://doi.org/10.1038/s43586-021-00055-w

Comments 2. In the Discussion section, please consider using the results of your findings with public health implications.  

Finally, I would like to commend the authors for their efforts to understand the complex interaction between social instability and isolation. Given the increasing aging population worldwide, prioritizing research in this area is essential for public health planning and policymaking.

Response 2: Thank you for your valuable comment. We have added content on the public health implications of our findings in the Discussion section.

Line 328: The identification of SI2 (personal communication), SF5 (lack of someone to talk to), and SI4 (belonging) as central nodes in the SF–SI network suggests clear entry points for public-health strategies. Routine dual screening for social frailty and isolation within primary-care settings—aligned with WHO’s ICOPE framework—can systematically flag at-risk individuals[50]. Building on this, nurse-facilitated interventions, such as peer companionship, intergenerational programs, and social prescribing to community assets (e.g., cultural, volunteer, exercise groups), may be particularly effective in restoring communication, confidant access, and belonging[51]. These models are highly compatible with China’s national priorities, including the Integrated Health and Elderly Care model and the Healthy China 2030 plan[52], both aiming to promote healthy ageing through community-based, person-centered services.

Reference:

  1. Huang, X., et al., From the WHO framework to integrated senior health and wellness hub program: an implementation journey. Front Public Health, 2025. 13: p. 1593490.
  2. Hu, L. and Y.W. Glavin, Integrating Health and Care for Older People in China: What Has Been Accomplished? What is Next? Int J Integr Care, 2023. 23(1): p. 16.
  3. Tan, X., X. Liu, and H. Shao, Healthy China 2030: A Vision for Health Care. Value in Health Regional Issues, 2017. 12: p. 112-114.

Round 2

Reviewer 2 Report

Comments and Suggestions for Authors

Thank you for your thoughtful revisions and detailed responses to the reviewers’ comments.
I have carefully reviewed the revised manuscript and find the changes satisfactory.
I appreciate the authors’ efforts in improving the quality of the paper.